# The Role of Artificial Intelligence in the Accurate Diagnosis and Treatment Planning of Non-Syndromic Supernumerary Teeth: A Case Report in a Six-Year-Old Boy

**DOI:** 10.3390/children10050839

**Published:** 2023-05-06

**Authors:** Rasa Mladenovic, Katarina Kalevski, Bojana Davidovic, Svjetlana Jankovic, Vladimir S. Todorovic, Miroslav Vasovic

**Affiliations:** 1Department for Dentistry, Faculty of Medical Sciences, University of Kragujevac, 34000 Kragujevac, Serbia; miki_vasovic@yahoo.com; 2Faculty of Stomatology, Pancevo, University Business Academy in Novi Sad, 26000 Novi Sad, Serbia; katarina.kalevski@sfp.rs; 3Department of Pediatric and Preventive Dentistry with Orthodontics, Faculty of Medicine, University of East Sarajevo, 73300 Foča, Bosnia and Herzegovina; bojana.davidovic@ues.rs.ba (B.D.); svjetlana.jankovic@ues.rs.ba (S.J.); 4School of Dental Medicine, University of Belgrade, 11000 Belgrade, Serbia; todent@yahoo.com

**Keywords:** hyperdontia, two supernumerary teeth, artificial intelligence

## Abstract

Hyperdontia can cause numerous aesthetic and functional problems. The diagnosis is made radiologically, and the most commonly used radiological method is orthopantomography, while CBCT is also used. CBCT has the advantage of being three-dimensional. Artificial Intelligence is widely used in medicine and dentistry, and it can create a specific algorithm to aid in diagnosis and suggest therapeutic procedures. In a case report, a 6-year-old boy was diagnosed with a supernumerary tooth between the upper central incisors. Orthopantomography revealed another impacted supernumerary tooth, and the patient was referred for CBCT. A platform for analyzing dental images, based on artificial intelligence, Diagnocat (Diagnocat Inc., San Francisco, CA, USA), was used for analysis and the AI system identified the supernumerary teeth and provided a complete plan for treatment. The use of AI in dentistry allows for faster and more accurate diagnosis and treatment planning.

## 1. Introduction

Hyperdontia, also known as supernumerary teeth, is a developmental dental anomaly that refers to the presence of excess dental structures that are not part of the normal dentition. Its prevalence ranges from 0.1% to 3.9%. Although hyperactivity of the dental lamina is the most widely accepted reason, the etiology of supernumerary teeth is multifactorial and can result from both genetic and environmental factors [1]. Supernumerary teeth are classified based on their morphology as either associated with specific syndromes or non-syndromic. They may occur as one or more teeth, unilaterally or bilaterally, with the most common location being between the two central maxillary incisors [1,2].

Hyperdontia in permanent dentition can cause numerous aesthetic and functional problems, such as crowding, median diastema, root resorption, ectopic eruption, cyst formation, and delayed tooth eruption [1,3]. Supernumerary teeth can also cause complications in the bones, which can endanger implant placement and exert pressure on nerves, leading to paresthesia and pain [4]. It is crucial for orthodontists, pediatricians, and general dentists to be knowledgeable about these teeth, especially since they typically work with children and can play a key role in detecting the condition early and devising a comprehensive treatment plan for the future [5].

Suspecting supernumerary teeth can be based on the presence of a larger median diastema or malposition of already erupted teeth, and the final diagnosis is made radiographically. Sometimes it is necessary to take multiple retroalveolar X-rays, especially in cases where teeth are interposed, to determine the exact position of the supernumerary tooth and its relationship with the permanent tooth germ. While conventional X-rays (including orthopantomography) can provide an initial evaluation, newer imaging techniques, such as CBCT, can offer a more precise understanding of the tooth’s exact position and spatial relationship with surrounding structures [6]. Today, in the diagnosis and treatment of supernumerary teeth, 3D segmentation models obtained through intraoral scanning are also used, as well as computerized tomography imaging [7,8]. The basic advantage of CBCT imaging is its three-dimensionality, and the fact that its exportable DICOM (Digital Imaging and Communications in Medicine) format can be used for numerous advanced options provided by digital dentistry. DICOM is a standard that facilitates the handling, storage, printing, and transmission of medical imaging information. It allows for the integration of imaging devices from various manufacturers and is now implemented by almost all imaging system manufacturers. By keeping all data in one file, DICOM enables the easy transmission of digital data sets over a network, without the need for human intervention. This eliminates the need for large archive systems that take up significant space and makes it easier for staff to sort and search for images in a digital archive. The use of web technologies also allows for easy access to images and findings from any computer with internet access, while ensuring security and confidentiality. In contrast, the STL (Standard Triangulated Language) format is used to describe the surfaces of bodies using triangles, with the density of triangles depending on the initial resolution and mathematical algorithms. While DICOM files provide more information about what is inside the volume, STL files provide more information about the surface of the volume [9].

Artificial Intelligence (AI) is a field of computer science that aims to perform tasks that typically require human intelligence. AI enables the automatic extraction of important features from input data for further interpretation of previously invisible patterns. The development of AI systems has gained momentum in many areas of medicine, and its use has become widespread in the healthcare sector. In dentistry, AI can create specific algorithms that aid in diagnosis and suggest therapeutic procedures [10].

Convolutional Neural Networks (CNNs) are a type of deep learning neural network commonly used in computer vision tasks such as image classification, object detection, and image segmentation. They are designed to automatically and adaptively learn spatial hierarchies of features from input images, without requiring manual feature extraction. The key building block of a CNN is the convolutional layer, which applies a set of filters (also known as kernels or weights) to the input image to produce a set of feature maps. Each filter acts as a feature detector, searching for specific patterns or features in the input image. During training, the CNN learns to adjust the filter weights to identify the most relevant features for a given task.

CNNs also typically include other layers such as pooling layers, which reduce the spatial dimensionality of the feature maps by subsampling, and fully connected layers, which perform classification based on the learned features. One of the key advantages of CNNs is their ability to learn hierarchical representations of the input data, with lower-level features learned in earlier layers and higher-level features learned in later layers. This allows the network to identify complex patterns and objects in images.

## 2. Case Report

A 6-year-old boy was referred to the Department for Dentistry, Faculty of Medical Sciences, University of Kragujevac, due to the presence of an atypical tooth in the region of central incisors. Clinical examination diagnosed a supernumerary tooth (CST—clinical visible supernumerary tooth) between teeth #11 and #21. An additional impacted supernumerary tooth (IST—impacted supernumerary tooth), between the upper central incisors, was observed on the orthopantomographic X-ray. To precisely localize and plan the therapy, the patient was referred for a CBCT scan.

### 2.1. Analysis

For the analysis of this case, we used Diagnocat (Diagnocat Inc, San Francisco, CA, USA), a platform for dental image analysis with the help of artificial intelligence (AI). This AI model is based on CNN. Data generation takes only a few minutes. For analysis and a complete report of an orthopantomographic X-ray, it takes approximately 2 min, while the analysis of a CBCT scan with 3D segmentation takes approximately 5 min (DICOM is necessary).

The first step in the analysis was the radiological report. The AI identified and numbered the teeth and recognized two supernumerary teeth (as shown in Figure 1).

The next step in the analysis was the endodontic report, where the AI provided an automatic selection of the best slice for each tooth and the selected region with the possibility of measurement and additional analysis (Figure 2). With these data, a good insight into the relationship between the root of the CST and the additional IST was gained. Based on the presented data, it was established that the IST-placed palatal would be the easiest to access, after the extraction of the CST through the alveolus (0.23 mm).

The most significant parts of this analysis are the three-dimensional segmentation and examination of structures through the implemented 3D Viewer (Figure 3A,B). The AI provides the option to select desired structures for segmentation or the entire scan. Within the viewer, an analysis of the relationship between the IST and the bone after the extraction of the CST was performed (Figure 3C), followed by an analysis of tooth #11 after the extraction of both supernumerary teeth (Figure 3D). Due to the small amount of bone structure that will remain after the extraction of both supernumerary teeth, the physiological mobility of tooth #11 is likely to be compromised, so immobilization with an immobilization splint is planned after the surgical intervention for additional stabilization.

### 2.2. Treatment

On the CBCT scan, no root resorption or associated pathological conditions were observed. The surgical intervention was performed under local anesthesia (2% lidocaine with 1:100,000 adrenaline) using an infiltration technique in the vestibule and then in the incisive papilla area. A sulcular incision was made with a #15 blade from one canine to the other, without any additional relaxing incisions in the vestibule. After lifting the mucoperiosteal flap, the CST was extracted, and access was gained to the IST through the extraction socket with osteotomy, followed by its extraction (Figure 4). The surgical wound was sutured with 4.0 silk suture.

After suturing, it was observed that tooth #11 had mobility that was beyond physiological limits, so an immobilization splint was fabricated which included teeth #11, #53, and #54 (Figure 5). The splint was removed after two weeks and, upon removal, tooth #11 had regained normal physiological mobility. Following a regular checkup, teeth #11 and #21 were found to be vital, and the patient was referred to continue orthodontic treatment.

## 3. Discussion

The prevalence of non-syndromic multiple supernumerary teeth is less than 1%, and their treatment depends on their type, position, and possible complications, which are determined clinically and radiographically. There is no clear consensus on the best time to remove unerupted supernumerary teeth [11]. Immediate surgery is commonly performed at around the age of 6 years, resulting in improved tooth alignment and a reduction in the need for orthodontic treatment. On the other hand, a delayed surgical approach at approximately 8 to 10 years of age, when the roots of adjacent permanent anterior teeth are complete, can avoid damaging roots that are still forming. The delayed approach is beneficial as the patient’s age can aid in the treatment process [12]. In our case report, we decided to perform surgical treatment after diagnosis due to the patient’s compromised aesthetics caused by the presence of a clinically visible supernumerary tooth and the failed eruption of the right lateral incisor.

Cone-beam computed tomography (CBCT) imaging is a valuable tool for diagnosing and planning the treatment of supernumerary teeth. In addition to diagnosis and treatment planning, CBCT imaging can also be used to monitor the progress of treatment and evaluate treatment outcomes. This is particularly important in cases where supernumerary teeth are associated with other dental or skeletal abnormalities, as successful treatment may require a multidisciplinary approach. Its ability to provide high-resolution, three-dimensional images allows clinicians to accurately diagnose and plan treatment for these challenging dental conditions, leading to better outcomes for patients. However, as with all medical imaging technologies, it should only be used when necessary and with appropriate precautions to minimize radiation exposure. It has diverse applications in different dental fields including implantology, orthodontics, periodontology, and restorative and forensic dentistry. With the aid of artificial intelligence algorithms, CBCT images can be analyzed, aiding healthcare providers in making diagnoses and clinical decisions [13].

Due to the difficulty, complexity, and length of the process for analyzing CBCT scans, we decided to use artificial intelligence, based on CNN, for the complete analysis and therapy plan in our case. Convolutional neural networks are a type of AI that are primarily utilized for object detection and segmentation tasks. CBCT scans provide three-dimensional reconstructions of the hard tissues of the head and neck, allowing dentists to visualize structures in all desired planes. Additionally, CBCT scans can identify clinically significant incidental findings in structures outside the dentist’s usual area of expertise [13].

The data obtained were extremely useful, and the surgical intervention followed the therapy plan created after the analysis with artificial intelligence. The recognition of supernumerary teeth and the precise, automated analysis by sections were of great importance for the final therapy plan.

A special emphasis was placed on three-dimensional segmentation, which provided numerous possibilities for analysis and treatment prediction. It is worth noting that STL files, representing oral structures, can be obtained from sources other than CBCT scans [14]. If STL files come from multiple sources, they can be integrated and used for the production of various modern design processes and tools for more precise and less invasive treatment. These benefits are used in navigational implantology, as well as guided endodontics, and can also be applied to the production of guides for the extraction of supernumerary teeth located in deeper bone tissue [15]. This further preserves bone tissue and shortens the duration of surgical intervention, which is particularly significant in younger patients. However, in our case, there was no need to create a guide, as the analysis provided insight into the position of the IST in the bone and its access through the alveolus of the CST, which was confirmed during the surgical intervention. As the removal of both supernumerary teeth weakened the right central incisor, a flexible splint was made to stabilize the tooth and prevent injury in the days following the operation [16].

Software tools based on artificial intelligence, such as Diagnocat, can automatically segment the CBCT file into separate anatomical structures, eliminating the need for manual segmentation. By providing personalized learning experiences, improving assessment tools, offering virtual simulations, and assisting in diagnosis and treatment planning, AI has the potential to transform the way dentistry is taught and practiced. Therefore, it is important to educate dental students on these possibilities and encourage early adoption of AI systems for the future daily work [17].

The limitation of the study is that despite the high accuracy and numerous benefits, AI tools cannot replace a dentist. Therefore, AI systems should remain as an aid in the diagnosis and treatment planning of supernumerary teeth.

## 4. Conclusions

Artificial intelligence, based on CBCT imaging, can accurately analyze data and provide the best approach for surgical treatment of supernumerary teeth. This greatly facilitates and accelerates the surgical procedure.

## Figures and Tables

**Figure 1 children-10-00839-f001:**
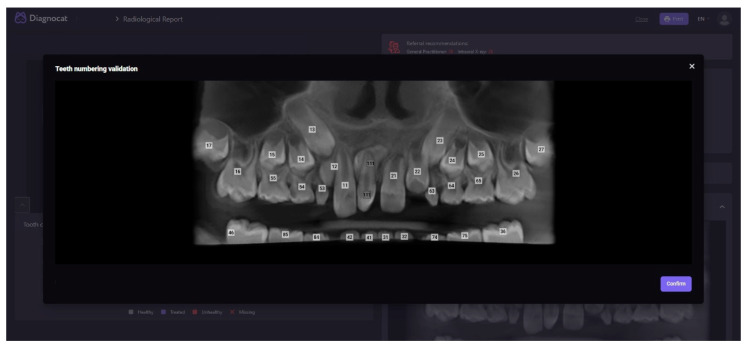
Automatic tooth recognition and numbering.

**Figure 2 children-10-00839-f002:**
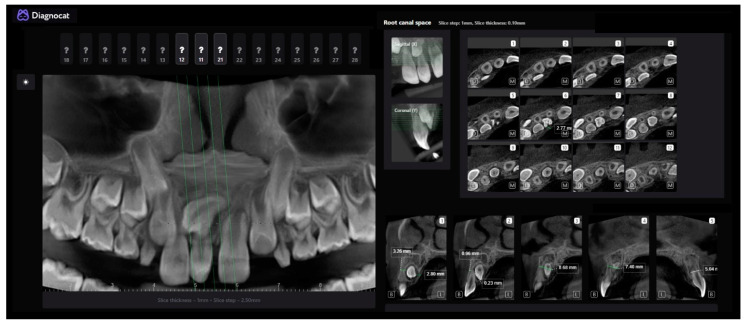
Detailed report and automatically selected slices.

**Figure 3 children-10-00839-f003:**
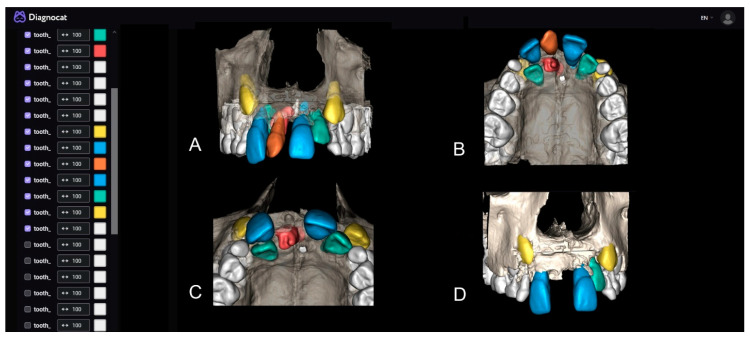
(**A**,**B**) STL segmentation and 3D visualization and simulation of the surgical procedure; (**C**) CST extraction; (**D**) Both supernumerary teeth extraction.

**Figure 4 children-10-00839-f004:**
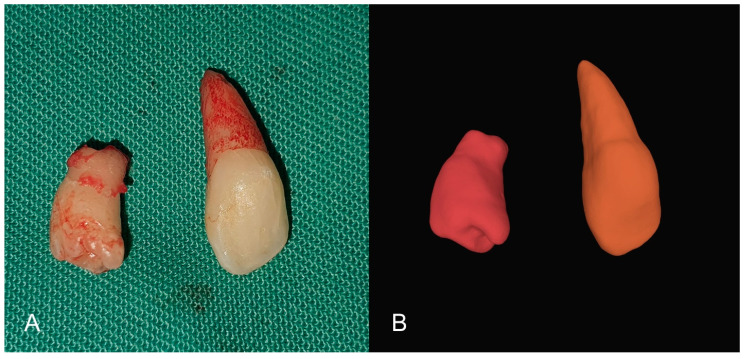
(**A**) Extracted supernumerary teeth; (**B**) STL segmentation of CBCT—clone teeth.

**Figure 5 children-10-00839-f005:**
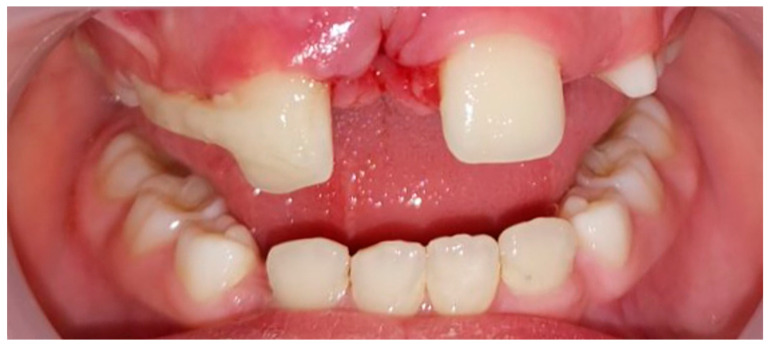
Immobilization splint.

## Data Availability

All data can be obtained from the corresponding author upon request.

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
