# Peer review of "The Role of Artificial Intelligence in the Accurate Diagnosis and Treatment Planning of Non-Syndromic Supernumerary Teeth: A Case Report in a Six-Year-Old Boy"

_children, 2023, doi:10.3390/children10050839_

Round 1

Reviewer 1 Report

This paper is a case report of a 6-year-old boy with a supernumerary tooth in the region of central incisors. The paper discusses the use of artificial intelligence (AI) based on convolutional neural networks (CNN) to analyze cone beam computed tomography (CBCT) scans and provide an accurate diagnosis and treatment plan. The AI system was able to identify and number teeth, select the best slice for each tooth, and provide a detailed report with measurements and additional analysis. The paper also discusses the use of three-dimensional segmentation and examination of structures through the implemented 3D Viewer, which provided numerous possibilities for analysis and treatment prediction. The paper concludes that the use of AI greatly facilitated and accelerated the surgical procedure for the removal of the supernumerary tooth.

Regarding title, you might consider better title "The Role of Artificial Intelligence in Accurate Diagnosis and Treatment Planning of Supernumerary Teeth: A Case Report in a Six-Year-Old Boy" or keep the original one.

The platform used for analyzing dental images mentioned in Abstract shall be more specified.

Your Introduction provides information that hyperdontia is a developmental dental anomaly that refers to any excess dental structure that is not part of normal dentition, and its prevalence ranges from 0.1% to 3.9%. It also mentions that the diagnosis of hyperdontia is made radiologically, and the most commonly used radiological method is orthopantomography, while CBCT is also used. Consider referencing that Adaptive Tooth Segmentation can be performed with AI even from intraoral scans in cases of hyperodontia (https://openreview.net/forum?id=O2DerS5oQ1 Model Adaptive Tooth Segmentation - Ruizhe Chen et al. )  Also computed tomography scans were used to assess hypodontia https://doi.org/10.1016/j.archoralbio.2023.105633 Sarbin Ranjitkar et al. 

You might consider introducing oher AI applications in dentistry-orthodontics also using diagnocat - for example that AI-Assisted CBCT Data Management in Modern Dental Practice is now shifting the paradigm doi.org/10.3390/electronics12071710 Renáta Urban et al

Reference Diagnocat (Diagnocat Inc, USA), properly. HQ city is missing.

The paper does not mention any limitations of using artificial intelligence in the treatment of supernumerary teeth. Please discuss them.

Diagnocat AI model is based on CNN. This case report discusses the use of artificial intelligence in the diagnosis and treatment planning of supernumerary teeth in a young patient which is fundamental part of orthodontics. The use of AI in dentistry now allows faster and more accurate diagnosis and treatment planning. This research may impact education in dentistry by highlighting the importance of incorporating AI technology into dental education and training. Dental students may need to learn how to use AI platforms for analyzing dental images and interpreting the results to provide the best treatment plan for their patients. Please consider discussing this aspect and referencing DOI 10.3390/educsci13020150 - Impact of Artificial Intelligence on Dental Education: A Review and Guide for Curriculum Update.

Additionally, this research may encourage further development and implementation of AI technology in dentistry, leading to more efficient and effective dental care in combination of dental monitoring and other tele-health solutions. Consider referencing DOI 10.3390/healthcare11050683 Strunga et al. - Artificial Intelligence Systems Assisting in the Assessment of the Course and Retention of Orthodontic Treatment

Please rewrite the Conclusion so it better summarises the findings presented in the paper. 

Otherwise your paper is useful case report and could be published.

Minor editing of English language required.

Author Response

Dear,
Thank you for your excellent and extremely useful comments, which will significantly improve the readability of our manuscript. We are pleased to have revised the manuscript in accordance with your suggestions.

COMMENTS:

Regarding title, you might consider better title "The Role of Artificial Intelligence in Accurate Diagnosis and Treatment Planning of Supernumerary Teeth: A Case Report in a Six-Year-Old Boy" or keep the original one.

            ANSWER: We like the suggestion of a new title, and we have accepted it.

The platform used for analyzing dental images mentioned in Abstract shall be more specified.

            ANSWER: We have corrected it in the text.

Your Introduction provides information that hyperdontia is a developmental dental anomaly that refers to any excess dental structure that is not part of normal dentition, and its prevalence ranges from 0.1% to 3.9%. It also mentions that the diagnosis of hyperdontia is made radiologically, and the most commonly used radiological method is orthopantomography, while CBCT is also used. Consider referencing that Adaptive Tooth Segmentation can be performed with AI even from intraoral scans in cases of hyperodontia (https://openreview.net/forum?id=O2DerS5oQ1 Model Adaptive Tooth Segmentation - Ruizhe Chen et al. )  Also computed tomography scans were used to assess hypodontia https://doi.org/10.1016/j.archoralbio.2023.105633 Sarbin Ranjitkar et al. 

            ANSWER: We have included the suggested studies in the text.

You might consider introducing oher AI applications in dentistry-orthodontics also using diagnocat - for example that AI-Assisted CBCT Data Management in Modern Dental Practice is now shifting the paradigm doi.org/10.3390/electronics12071710 Renáta Urban et al

            ANSWER: Included in the revision.

Reference Diagnocat (Diagnocat Inc, USA), properly. HQ city is missing.

            ANSWER: We have corrected it in the text.

The paper does not mention any limitations of using artificial intelligence in the treatment of supernumerary teeth. Please discuss them.

            ANSWER: The last paragraph has been added to the discussion about the limitations.

Diagnocat AI model is based on CNN. This case report discusses the use of artificial intelligence in the diagnosis and treatment planning of supernumerary teeth in a young patient which is fundamental part of orthodontics. The use of AI in dentistry now allows faster and more accurate diagnosis and treatment planning. This research may impact education in dentistry by highlighting the importance of incorporating AI technology into dental education and training. Dental students may need to learn how to use AI platforms for analyzing dental images and interpreting the results to provide the best treatment plan for their patients. Please consider discussing this aspect and referencing DOI 10.3390/educsci13020150 - Impact of Artificial Intelligence on Dental Education: A Review and Guide for Curriculum Update.

            ANSWER: Done in the discussion.

Additionally, this research may encourage further development and implementation of AI technology in dentistry, leading to more efficient and effective dental care in combination of dental monitoring and other tele-health solutions. Consider referencing DOI 10.3390/healthcare11050683 Strunga et al. - Artificial Intelligence Systems Assisting in the Assessment of the Course and Retention of Orthodontic Treatment

            ANSWER: Done in the discussion.

Please rewrite the Conclusion so it better summarises the findings presented in the paper. 

ANSWER: We have revised the conclusion.

Reviewer 2 Report

In this paper, the authors present a clinical case of hyperdontia in a 6-year-old boy and its removal.

Five original pictures are listed in the manuscript. The discussion is very scarce, and the list of literature consists of only 10 references.

I suggest:

1. expand the Discussion chapter

2. increase the number of references (three recent references are attached)

3. English needs to be improved by native speakers.

Akitomo T, Asao Y, Iwamoto Y, Kusaka S, Usuda M, Kametani M, Ando T, Sakamoto S, Mitsuhata C, Kajiya M, Kozai K, Nomura R. A Third Supernumerary Tooth Occurring in the Same Region: A Case Report. Dent J (Basel). 2023 Feb 12;11(2):49. doi: 10.3390/dj11020049. PMID: 36826194; PMCID: PMC9955779.

Hajmohammadi E, Najirad S, Mikaeili H, Kamran A. Epidemiology of Supernumerary Teeth in 5000 Radiography Films: Investigation of Patients Referring to the Clinics of Ardabil in 2015-2020. Int J Dent. 2021 Feb 22; 2021:6669436. doi: 10.1155/2021/6669436. PMID: 33688347; PMCID: PMC7925020.

Ahammed H, Seema T, Deepak C, Ashish J. Surgical Management of Impacted Supernumerary Tooth: A Case Series. Int J Clin Pediatr Dent. 2021 Sep-Oct;14(5):726-729. doi: 10.5005/jp-journals-10005-2008. PMID: 34934291; PMCID: PMC8645632.

In this paper, the authors present a clinical case of hyperdontia in a 6-year-old boy and its removal.

Five original pictures are listed in the manuscript. The discussion is very scarce, and the list of literature consists of only 10 references.

I suggest:

1. expand the Discussion chapter

2. increase the number of references (three recent references are attached)

3. English needs to be improved by native speakers.

Akitomo T, Asao Y, Iwamoto Y, Kusaka S, Usuda M, Kametani M, Ando T, Sakamoto S, Mitsuhata C, Kajiya M, Kozai K, Nomura R. A Third Supernumerary Tooth Occurring in the Same Region: A Case Report. Dent J (Basel). 2023 Feb 12;11(2):49. doi: 10.3390/dj11020049. PMID: 36826194; PMCID: PMC9955779.

Hajmohammadi E, Najirad S, Mikaeili H, Kamran A. Epidemiology of Supernumerary Teeth in 5000 Radiography Films: Investigation of Patients Referring to the Clinics of Ardabil in 2015-2020. Int J Dent. 2021 Feb 22; 2021:6669436. doi: 10.1155/2021/6669436. PMID: 33688347; PMCID: PMC7925020.

Ahammed H, Seema T, Deepak C, Ashish J. Surgical Management of Impacted Supernumerary Tooth: A Case Series. Int J Clin Pediatr Dent. 2021 Sep-Oct;14(5):726-729. doi: 10.5005/jp-journals-10005-2008. PMID: 34934291; PMCID: PMC8645632.

Author Response

Dear,
Thank you for your valuable comments. We are pleased to have revised the manuscript in accordance with your suggestions.

COMMENTS:

  1. expand the Discussion chapter

            ANSWER: Our discussion is now expanded.

  1. increase the number of references (three recent references are attached)

            ANSWER: The suggested studies have been implemented in the revised manuscript.

  1. English needs to be improved by native speakers.

            ANSWER: The entire manuscript has undergone English language editing.